# Volunteering in the Citizen Science Project “Insects of Saxony”—The Larger the Island of Knowledge, the Longer the Bank of Questions

**DOI:** 10.3390/insects12030262

**Published:** 2021-03-20

**Authors:** Nicola Moczek, Matthias Nuss, Jana Katharina Köhler

**Affiliations:** 1PSY: PLAN Institute for Architectural and Environmental Psychology, 10245 Berlin, Germany; 2Museum für Naturkunde, Programme Society and Nature, 10115 Berlin, Germany; 3Senckenberg Natural History Collections Dresden, Museum of Zoology, Lepitoptera Section, 01109 Dresden, Germany; matthias.nuss@senckenberg.de; 4Department of Cognition, Emotion, and Methods in Psychology, University of Vienna, 1010 Vienna, Austria; janakatharina.koehler@gmail.com

**Keywords:** social psychology, citizen science, participant demographics, motivations, organisational framework, volunteer management

## Abstract

**Simple Summary:**

Motivations and the organisational frame for volunteering in citizen science projects remain understudied. This study applied a new measurement inventory called Motivational and ORganisational Functions of voluntary ENgagement in Citizen Science (MORFEN-CS), by which established findings of psychology on volunteering in social situations were transferred to volunteering in biodiversity and environmental sciences. It was applied in the citizen science project Insects of Saxony. The results are presented in this article and implications for measuring motivations and for project and volunteer management in citizen science projects are discussed.

**Abstract:**

In a cross-sectional survey study (*N* = 116), volunteers of the project Insects of Saxony were asked about their current and past volunteering activities, their motivations, their rating of organisational offers, their knowledge, their satisfaction with the project and their personal contribution, and their intended future involvement. Participants in the study were mostly male, well-educated, over 50 years old, and had been volunteering in biodiversity projects for a long time. They were driven by both pro-social (altruistic) and self-serving (egoistic) motivations, but rated the pro-social functions as more important for their engagement. Communication and feedback were rated the most important organisational offers. Participants also reported a knowledge increase during project participation. While the volunteers were satisfied with the overall project, they were significantly less content with their own contribution. Results from the survey were followed up with a group discussion (*N* = 60). The anecdotes revealed the participants’ regret of not having more time for their hobby, and they emphasised the challenges that arise from the different scientific approaches of the various disciplines. Most participants indicated that they want to continue their volunteering. Implications for measuring motivations in citizen science projects and for volunteer management are discussed.

## 1. Introduction

In the past ten years, the active participation of citizens in scientific research processes has experienced a considerable boom [1]. Half of the estimated 1.2 million participants in 174 European citizen science projects primarily collect data, and an additional 27 percent are also involved in project design and data analysis [2].

Covering a broad range from all disciplines, most projects still focus on natural sciences, including biodiversity and environmental sciences. Especially in these projects, the involvement of volunteers (hereafter *citizen scientists* or CSs) is well established, and they not only contribute data but also help to implement effective conservation measures [3]. In particular, insect diversity and its function in ecosystems have become an important research field [4,5]. This is linked to the *Convention on Biological Diversity* (CBD) [6], which obliges nations to develop effective strategies for conservation and monitoring. In accordance with the CBD, the German Federal Government implemented several programmes such as the *Action Programme for Insect Protection*, started in September 2019 [7]. Data gathered in this programme are also valuable for the *Red List of Threatened Species* [8,9]. How urgently the topic needs our attention was shown impressively in a study carried out by German CSs. They had been examining insect populations in a total of 63 areas and found that since 1989 the total mass of flying insects has decreased by more than 75% [10].

This type of monitoring requires the help of many experts for very different species, who regularly collect data on a large scale at many different locations. In Germany, as early as 2007, the Federal Ministry for the Environment, Nature Conservation, and Nuclear Safety named this kind of participation *citizen science* [6]. Currently, in Germany there are many initiatives which support data acquisition and data availability on insects [8].

What drives people to contribute actively to a scientific research project in their free time, and which organisational aspects enable and support this volunteering? Very little empirical data are available regarding the reasons driving people to actively contribute to a scientific research project in their free time [3,11], as well as regarding the organisational aspects that enable and support these volunteering actions. Meanwhile, the use of different survey instruments and methods makes the available findings difficult to integrate [12].

Consequently, our study will apply a new instrument to measure the motivational and organisational functions of volunteering for citizen science. In doing so, the study contributes to the body of knowledge around environmental volunteering, as well as to the scientific tool kit for studying participation.

### 1.1. A Social Psychology Perspective on Volunteering

When, why and under which circumstances people act *pro-socially* has long been a key question of social psychology. For over two thousand years there has been a debate about whether the drivers for social action are *egoistic* or *altruistic*. “The distinction between altruism and egoism is qualitative, not quantitative; it is the ultimate goal, not the strength of the motive, that distinguishes altruistic from egoistic motivation”, Batson and Shaw stated 1991 [13]. In their review, they provided evidence of altruistic motivation for helping behaviour. Specifically, they found evidence for a mechanism where empathy can lead to altruism (*empathy-altruism hypothesis*), and altruism can lead to *prosocial behaviour*. Consequently, they advocated a pluralistic model to explain prosocial behaviour, which considers both egoistic and altruistic motivations [13]. In addition, the *functional approach* describes that the same volunteering can be based on different motivations and that prosocial behaviour can fulfil more than one function for individuals. The term was coined in 1956 by Smith et al. [14] and 1960 by Katz [15] when measuring attitudes, and transferred to motivational research in 1998 by Clary, et al. [16]. They described six functions which they combined in their Volunteering Function Inventory (VFI). The functions comprise *values*, i.e., expressing values through volunteering, *understanding*, i.e., learning new things, *social*, i.e., social relationships, *career*, i.e., furthering one’s career, *protective*, i.e., protecting oneself from emotions like guilt, and *enhancement*, i.e., enhancing positive effect. Interestingly, these six functions are more or less egoistic in nature.

Most studies on the subject had looked at *spontaneous*, *short-term helping behaviour in social contexts*. Penner was in 2002 the first to focus on *sustained volunteerism*, which he defined as “long-term, planned, prosocial behaviours that benefit strangers and occur within an organisational setting”. He found evidence for a multitude of factors that influence sustained volunteering, such as demographic factors, situational factors, and factors related to the organisations, which he transferred to his *Conceptual Model of the Causes of Sustained Volunteerism* [17] (see Figure 1).

In this article, knowledge from social psychology is applied to activities in a citizen science project focussing on wild animals and their habitats. Based on the findings from psychological research on social volunteerism, it was postulated that volunteering for natural science projects is also influenced by altruistic and egoistic motives, as well as organisational and personal factors.

### 1.2. Motivations to Volunteer in Biodiversity and Environmental Sciences

Little empirical data are available on the motivational structures and the organisational framework that promotes sustained volunteering in citizen science projects [3,11]. Where there are data, studies produce mixed findings on whether egoistic or altruistic motivations are more important for participation. Studies considering altruistic motivation to be more important found that their participants primarily want to help the environment [19], or take care of the environment [20]. Studies that found egoistic motivations to be more important showed that their participants want to learn more about the topic [21], or have personal interest in it [22]. There is also evidence for differences in motivation over time. While *starting* to volunteer is more driven by egoistic motives (e.g., personal interest), *sustaining* the engagement requires both egoistic and altruistic motives [23].

### 1.3. Measuring Motives for Volunteering in Biodiversity and Environmental Sciences

Studies investigating motives for volunteering apply a multitude of different methods, such as interviews or focus groups, quantitative questionnaires, or combinations of both. The inventories available, like the VFI [16] and the Attitude Structure of Volunteers (SEEH) [24], focus on charity volunteering. Their items build on principles of social relationships, such as reciprocity, that do not apply to the natural environment or animals in the same way. Furthermore, they mainly measure egoistic motives. The lack of a standard inventory makes the findings of existing studies difficult to compare and integrate [12,18]. Another possible source of variation in the literature on CSs motivations is the high degree of diversity among the projects. Projects vary strongly with regard to their methodological approach (e.g., mass participation or select trained volunteers), or their complexity (for typologies see [1,25,26]).

To improve this, a psychometric scale system called Motivational and ORganisational Functions of voluntary ENgagement in Citizen Science (MORFEN-CS) was developed, tested and validated between 2016 and 2018 by Moczek [18]. MORFEN-CS was based on the theoretical Model of Influences on Participation in Citizen Science [11,27], and on Penner’s Model Causes of Sustained Volunteerism, [17], and utilised in parts the VFI [16] and the SEEH [17]. The development of new items followed the approach of classical test theory (CTT) [28], where initial items were compiled through literature research, then refined and complemented in four focus group discussions (study 1, *N* = 38) and expert interviews [18]. This approach is one way of ensuring construct validity. Several items were formulated for each characteristic to be measured (e.g., the function *Social Motives*). The items must all represent the characteristics to be measured appropriately and comprehensively in terms of content, and they must also be formulated clearly and comprehensibly. The items should represent as many degrees of expression of the characteristic to be measured as possible—i.e., low, medium and high characteristic expressions (measured with P_i_: item difficulty indices). Each item should as clearly as possible separate persons with a strong characteristic from persons with a weaker characteristic (measured with r_it_, item-total correlation). The number and order of the items should be determined in such a way that the scale can be easily processed by the respondents and at the same time its psychometric quality criteria (especially reliability, in this case tau-equivalent reliability or Alpha, and validity) optimised.

MORFEN-CS was first applied in a pre-test (Study 2, *N* = 11) and in an online-survey using a four-level scale (study 3, *N* = 209). Respondents of Study 3 were German CSs who had collected hair samples of the European Wildcat over several years. These were genetically analysed and transferred to a nationwide database [29]. With Study 3, a first item analysis, construct validation and confirmatory factor analysis (CFA) were conducted. The resulting model included eight motivational functions loading on two superior factors (four *prosocial or altruistic*, and four *self-serving or egoistic functions*), and four organisational functions loading on one superior factor [18].

Before being applied in Study 4, the items were revised on the basis of the item analysis to further increase statistical reliability. Some project-specific items of Study 3 were replaced by more general ones. To increase statistical variance, a six-level response scale was presented. Participants in this survey were volunteers from two citizen science projects in Saxony, one of which is *Insects of Saxony*. The second project explores the effects of insect-friendly habitats on diversity [30] (Study 4, together *N* = 216). The multi-factorial structure of MORFEN-CS was replicated by confirmatory factor analyses. Once again, the eight motivational and four organisational functions were found. In sum, there is evidence that different functions of volunteering can be measured with MORFEN-CS. For the resulting factor solutions of MORFEN-CS obtained in these previous studies see the Appendix A).

### 1.4. Research on the Citizen Science Project Insects of Saxony

We will start by introducing the German project *Insects of Saxony*. As mentioned above, this was one of the two groups participating in the validation study 4.

The *Entomology Working Group Saxony* of Naturschutzbund Germany (NABU) was founded in 2006 by thirteen volunteer members (only one scientist). They dedicated themselves to researching and protecting the highly complex and diverse insect world, in particular to identifying and monitoring the approximately 25,000 insect species occurring in Saxony. Since spring 2011, they have been operating an open access internet platform and database *Insects of Saxony* [31]. In 2016, the *Senckenberg Society for Nature Research* joined as a co-operation partner with a module for the collection of historical data from scientific collections as well as the literature and diary entries. Since then, a mobile app has been available. Since 2019, the app has allowed recording of all insects. It contains photos and descriptions for >450 insect species. The group organises two workshops and a yearly excursion of several days, as well as editing the *Saxon Entomological Journal* [32].

*Insects of Saxony* does not provide specified or defined tasks, like focusing on a given selection of insect species, nor clear instructions for the volunteers. This differentiates this project from many other citizen science formats using a more task-led approach [4]. CSs are free to choose on what insect species they specialise and when, where, how, how often they do their research. Outcomes are scientific findings and individual learning of the participants [26]. Some CSs are specialised in a certain region, trying to record as many species as possible, while others focus on a certain group of insects. Following the framework for *Public Participation in Scientific Research* (PPSR) [26], CSs in this project can take on several roles according to their interest and level of expertise: (a) as *contributors* (primarily of data), (b) as *collaborators* (they contribute data, but also support project design, data analysis, and/or dissemination of results), (c) as *co-creators* (the research question is generated by scientists and volunteers together, and volunteers are actively involved in most or all aspects of research) and (d) as *colleagues* (CSs independently conduct research that advances knowledge in a scientific discipline). There were 202 active platform members in 2018, 247 during 2019, 432 in 2020. In total 942 members have registered since project start.

To ensure data quality, every reported finding is verified by an expert team for accuracy and plausibility. This also includes the technical requirements for species identification. Some can be identified in the field by observation without any technical equipment, for some magnification is needed either by magnification lenses or macro-photography, and few need to be examined with a stereomicroscope or methods such as DNA analysis or statistical analysis of morphometric measurements. Only proofed data are published on the platform, showing the records in an interactive map, the dates in a phenology diagram, and the photos along with the species descriptions. By November 2020, there were >330,000 records for >7000 species and 86,000 photos for 4500 species. Periodically, the data are made available at *Global Biodiversity Information Facility (GBIF)* [33]. At times, there are calls for looking at certain species in particular, and then detailed documentations of their occurrences in Saxony are published separately, e.g., for the glow worm *Lamprohiza splendidula* [34] and the large wood bee (*Xyloscopa violacea*) [35]. The project has been very successful so far, considering the high number of active volunteers, and the expansive record of species in the open access database.

### 1.5. Research Questions

This article extends knowledge from social psychology on volunteering in social situations to volunteering in natural science projects. The following research questions guided us:What are the main personal circumstances and demographics of the citizen scientists?Which personal motives can be differentiated?Which organisational structures and offers are rated important for the participants’ commitment?Is there an increase in knowledge through voluntary work?How contented are the citizen scientists with the overall project and their personal contribution to the project?Do they want to continue their volunteering in the project?

Questions 1–3 and 6 were derived from the *Model of influences on participation in citizen science projects* [11] (see Figure 1). While the model depicts the process of engagement over time, we were constrained to a cross-sectional survey study during ongoing engagement. Questions 4 and 5 allow us to assess desired project outcomes (learning and acquiring new knowledge; e.g., [26]) and the valuation of the volunteering by the CSs.

## 2. Materials and Methods

The questionnaire was provided between 27 November and 15 December 2018 via an online survey tool [36] to 202 volunteers. For demographics, participants indicated their age, gender, highest educational qualification, employment status, and the size of their place of residence. For their engagement, participants indicated the duration (seven categories: <1 year, 1–2 years, 3–5, 6–10, 11–20, 21–30, >30 years) and frequency of previous activities in nature projects (8 items, see Table 1; closed answer format “never”, “occasionally”, “often”), the scope of their engagement in the current project (average weekly working hours in 2018). They completed a self-assessment of their knowledge before the start of the volunteering and at the time of the survey (five categories; laity, low, medium, or extensive expertise, expert), an overall assessment of their contentment with the project and with their personal contribution (11-point scale from 0%, 1–10%, up to 91–100%), and indicated their intended future involvement with the project (five statements, six-point polarity profile ranging from 1 = *stop investing time* to 6 = *invest much more time*). The 42 items from MORFEN-CS (see Section 1.3.) were rated on a six-point scale (1 = *does not apply at all* to 6 = *fully applies*). For a full list of items included in MORFEN-CS, both in German and English, see the Appendix A).

In order to evaluate the statistical construct and criterion validity of MORFEN-CS, additional questions and scales were used, but they are not presented in this article, see [18]. Participants gave informed consent, were fully debriefed about data management and the purpose of the study and were invited to a presentation and discussion of the results.

Only datasets with a finished page 5 (out of 8) and less than 20% missing values were included in the analysis. The net sample consisted of 116 questionnaires (57.4%). Analyses were carried out with Excel (version 15.29.1), RStudio (version 3.3.2) and R-package lavaan [37].

The results of the survey were followed up with a qualitative discussion on 30th March 2019, at the Senckenberg Museum of Zoology in Dresden, during the annual workshop. About sixty people participated, of whom half had completed the online questionnaire. After presenting the main results by the first author we focussed in the discussion on the volunteers’ satisfaction with their own engagement. The many and lively reports were noted down and later categorised by the authors unsystematically, as they were only exploratory in nature.

## 3. Results

First, we will present the results from the survey, then from the group discussion. The structure will follow the research questions.

### 3.1. What Are the Main Personal Circumstances and Demographics of the Citizen Scientists?

Of the 116 respondents, 71.6% are men and 26.7% women (1.7% did not specify). On average they were 52.3 years old (*SD* = 13.9). Nearly 70% have an academic degree (Bachelor, Master, PhD). A total of 8% are still at university, 24% completed vocational training. Most of the respondents are employed (21–40 h per week 34.8%, >40 h 24.3%), 24.3% are on pension or in retirement, and all reside in Saxony (Germany). A total of 33% live in one of the two cities of Dresden and Leipzig, which have more than 500,000 inhabitants each. A total of 30% live in places with fewer than 10,000 inhabitants. Nearly 11% have been active in the project since the beginning in 2010. A total of 15% joined in 2016, 13% in 2017 and 23% in 2018. On average, respondents have been active for 43.6 months (*SD* = 33). A total of 57% volunteered one hour/week, 17% up to 2 h and 4.5% up to 3 h. Almost 10% spend more than 6 h/week. Together they spent 11,605 h volunteering in 2018.

The majority have been active in biodiversity for a long time, with 37.9% for more than 30 years, 12.6% between 21 and 30, 16.8% between 11 and 20, 13.7% between 6 and 10, 12.6% between 3 and 5, and 4.2% between 1 and 2 years. Taking a closer look at the activity profile of the CSs, 95% often or frequently carry out systematic scientific observations. A total of 94% of them publish their findings on the open access database (occasionally or often). Most of them do not measure or analyse data, nor do they formulate new research questions. Only a few are involved in the quality checking of the reports, as this is reserved for CSs who are qualified as experts (Table 1).

### 3.2. Which Personal Motives Can Be Differentiated?

Looking at the function *citizen science* (see Table 2), item 5 “help to stop the loss of habitats of insects” was rated highest (*M* = 5.63, *SD* = 0.69) and item 15 “contribute to species identification and environmental monitoring” as the second highest (*M* = 5.61, *SD* = 0.79), followed by other items from the function nature conservation values (see Table 2). Overall, the five items relating to nature conservation values were rated very high (*M* = 5.43, *SD* = 0.73). All five items show power indices between P*_i_* 83.9 and 92.2, indicating that they were easy to answer for the respondents of the survey. Because the item-total correlation was within the requested range (r*_it_* = 0.4–0.7), they differentiate well between subjects with extreme characteristic values [38]. Note that all items and their means in descending order can be found in the Appendix A).

After items 5 and 15, *socio-political responsibility* (*M* = 4.77, *SD* = 1.14), *citizen science* (*M* = 4.47, *SD* = 1.19) and *social motives* (*M* = 4.71, *SD* = 1.54) were rated most important for commitment to the project. These four groups of motivational functions are provisionally called *pro-social* functions (or altruistic; marked green in Figure 2).

The four *self-serving* motivational functions (or egoistic, provisionally; marked blue in Figure 2) were ranked much lower, such as *enhancement* (*M* = 3.28, *SD* = 1.35), *work life balance* (*M* = 3.28, *SD* = 1.63), and *career* (*M* = 2.58, *SD* = 1.81). With one exception: *Qualification* (*M* = 4.34, *SD* = 1.33), which represents self-organised learning, which was ranked at place 5.

### 3.3. Which Organisational Structures and Offers Are Rated as Important for Commitment?

Three of the four *organisational* functions (marked orange and OR in Figure 2) were also ranked as relevant motives, starting with *communication and feedback* (*M* = 4.28, *SD* = 1.18), *organisation* (*M* = 4.11, *SD* = 1.32), and *coordination* (*M* = 3.91, *SD* = 1.31), whereas *training* was rated as less relevant (*M* = 3.19, *SD* = 1.30).

### 3.4. Is There an Increase in Knowledge through Voluntary Work?

The self-assessments of knowledge showed an increase from project start to the time of the survey (Table 3). A total of 36.9% stated to have had lay knowledge prior to starting, 33% low, 25.2% medium, 27.8% extensive expertise and 9.6% expert knowledge. The self-reported expertise at the time of the survey was estimated to be much higher: lay knowledge is 9.6%, low 9.57%, medium 28.7%, extensive expertise 38.3% and 15.7% expert knowledge. The effect is highly significant (*t* (114) = 8.786, *p* < 0.001, *d* = 0.508 (medium effect size), 95% CI for *d* (0.387, 0.629); single sample *t*-test, one-sided).

### 3.5. How Satisfied Are the Citizen Scientists with the Overall Project and Their Personal Contribution to the Project?

While contentment with the project was high (*M* = 79.3%, *SD* = 17.7; see Figure 3), satisfaction with one’s own contribution to the project was rated significantly lower (54.5%, *SD* = 25.4; *t* (113) = −9.918, *p* < 0.001, *d* = −1.121 (large effect size), 95% CI for *d* (−1.406, −0.837); single sample *t*-test, one-sided).

This result was therefore a main topic in the group discussion. Most of the participants attributed their dissatisfaction to the lack of time: “I actually want to do more, I plan to do that and then it often doesn’t work”. Some reported different technical problems with the app so they could not log their finds properly. Because the number of participants and reports grew so fast, one of the voluntary experts involved in the quality checking of the data mentioned: “I no longer have the opportunity to enter my own observations, because I need all my free time to check other data. We need to have 10 to 12 people for quality checks now”. Some legal issues were discussed, too: “During my excursions I often do not have the right to access the respective areas. I have therefore drawn up a list of landholders, huntsmen and foresters, which was time-consuming. I always ask them before I do research in their areas. It is difficult to obtain such permits”. Moreover, co-operation challenges were mentioned: “It is about the interplay of different areas of knowledge between plant and insect scientists and about different methods such as photography, microscopy, research, digital applications”. Another participant added: “There is a close link between scientists and lay people in this project, but we need to find a common language, so that we can advance research. This could be a pilot project”. The closing word of the discussion was: “Dissatisfaction is my motor, I still want to do it better! The size of the task appeals to me: The larger the island of knowledge, the longer the bank of questions”.

### 3.6. Do the CSs Want to Continue Their Volunteering in the Project?

For 2019, 28% indicate that they want to spend the same amount of volunteering time as in 2018. A total of 3.5% plan to reduce their time (cumulative values 1–3), 68.1% plan to spend (much) more time (cumulative values 4–6). A total of 82% would like to know more about the habitats of insects, 79.6% wanted to learn more about individual insects and 80.5% more about different insects. A total of 78.3% aim to contribute to the protection of insects (see Table 4). No statistical correlation was found between the items of the intended participation and the items of the organisational functions.

## 4. Discussion

### 4.1. Summary of Results

The study contributes to our understanding of the motivational and organisational functions of volunteering in biodiversity and environmental sciences projects. The findings expand current knowledge and contribute to the tool kit available for studying participation. This is achieved by applying the new scale system MORFEN-CS.

Our survey showed that most of the participants are male, well-educated, and over 50 years old. These profiles perfectly match many other citizen science studies [39]. Most of the respondents spend their volunteering time with systematic, scientific observations of insects and they publish their findings on the online database, thereby taking on the roles of *contributors* or *collaborators* [26].

We found that multiple motivational and organisational functions are relevant to their volunteering (research questions 2 and 3). All four *pro-social* (or altruistic) functions were rated very high (Table 2, Items 1–16). The functions of *nature conservation values* and *socio-political responsibility* received the highest average agreement, indicating that volunteers want to make contributions to nature conservation that are socially and politically meaningful. The motivational functions *citizen science* and *social motives* were also rated as (very) important for the volunteering. In other words, the CSs are looking to support and be part of a community which shares a common research interest. They want to make contact with other like-minded scientists and volunteers to better understand scientific processes and engage in knowledge exchange. These findings parallel those of other studies, which found similar functions of volunteering (e.g., [40,41]). The four *self-serving* (or egoistic) motivational functions, such as *enhancement*, *work life balance* and *career* (Table 2, items 17–28), were rated (much) lower. Since the majority of the citizen scientists are already over 50 years old and probably have established careers, it is not surprising that improving their skills is not an important function of volunteering for them. In sum, these findings are in line with studies that found multiple different functions to be important for both social volunteering [16,24], and engagement in citizen science projects [19,20,21,22,23]. In addition, our findings support other studies that found altruistic functions to be more important [19,20]. Three of the four *organisational* functions, i.e., *communication and feedback*, *project-organisation*, and *-co-ordination*, were also rated as relevant for the engagement. This is especially true for participants for whom it is important that their own contribution is relevant to the project (item 38), that the project is well-organised (item 39), and that they can independently determine time and duration of their engagement (item 32). Particularly the high rating of item 38 is in line with a review which showed that citizen scientists highly value the open communication of project findings [42]. Another study also found the importance of feedback, in particular on learning outcomes [43]. Interestingly, the participants rated the two functions relating to learning and knowledge very differently: *qualification* (high rating; *self-serving function*) and *training* (low rating; *organisational function*). In other words, participants indicated having acquired new skills through the project, despite receiving little training. This matches the project structure, which stipulates some training (e.g., excursions, lectures), but mostly relies on self-directed learning. This maps well onto self-determination theory (SDT) [44] which postulates autonomy as one of three central human needs and an important component of motivation. In the Wildcat-study (see introduction), learning was organised more to be top-down, and opposite ratings of *training* and *qualification* were found [18]. That project followed a task-oriented approach, in which training in the application of scientific methods of data collection was a prerequisite for participation [29]. This exemplifies also the importance of organisational functions. The high rating of the *qualification function* is also in line with the self-assessed–significant knowledge increase from project start to the time of the survey on expertise on insects and their habitats in Saxony (medium effect size, research question 3). Similarly, a review article found increases in both general and project-specific knowledge through participation in citizen science projects [45]. One result was particularly surprising: while the contentment with the overall project was high on average, participants’ contentment with their own contribution was significantly lower (large effect size, research question 5). Anecdotes from the group discussions revealed that volunteers regret not having more time for their hobby, and they emphasize the challenges of the scientific approach of different disciplines. The survey also confirmed that the CSs plan to continue their volunteering (research question 6). Almost one-third plan to spend the same amount of time as in 2018, two-thirds plan to spend (much) more time; most of them want to gain more knowledge about insects and their habitats. This corroborates findings from another study, where participants of an entomological citizen science project had a more positive attitude towards insects after project participation [46].

### 4.2. Limitations

Despite a high response rate, our sample is relatively small and homogeneous with the risk of self-selection bias. During the discussion at the workshop, we received feedback from some that, as a matter of principle, they never take part in surveys. A few of the participants explicitly regretted this in retrospect, as the presentation and discussion of the findings had shown them how valuable such studies can be for the project managers and also their own engagement. It is also conceivable that people tend to take part in such a survey who were more likely to be driven by pro-social motivations. All of this can lead to a bias in the respondent pool and therefore also in the results.

In his *Conceptual Model of the Causes of Sustained Volunteerism*, Penner models the influences on volunteerism over time [17]. Likewise, there is evidence that the functions of volunteering in biodiversity and environmental sciences differ over time [11,23]. As we were constrained to a cross-sectional survey design, this study cannot provide any insights into functions of volunteering over time. Future studies could apply MORFEN-CS in longitudinal study designs to explore temporal effects.

### 4.3. Outlook and Recommendations

With MORFEN-CS, Moczek transferred established findings of psychology on volunteering in social situations [16,17] to volunteering in biodiversity and environmental sciences. Applying this new instrument, we aimed to contribute to the development of tools available for studying participation, and to enhance the comparability of research findings between projects and participation formats. MORFEN-CS is not to be understood as a finished inventory, but rather as a starting point for future research that is open for constructive development. Future research could adapt, apply, and develop MORFEN-CS also across contexts in order to compare different projects and people involved. Since these scales were developed for collective diagnostics, some low overall alpha-values are considered acceptable, and the sub-scales confirm the multidimensionality. The factor structure should be further explored with larger samples, given some conflicting findings on the second-order factors (altruistic vs. egoistic) [18]. In addition, an abridged version of MORFEN-CS was developed, which contains only 12 items. This is currently still in the testing phase.

Future research should also focus on the effects of collaboration between institutional scientists and volunteers [47], or on project staff like trainers and volunteers e.g., [48] especially regarding the effects on public understanding of science [49]. As our study is only a first step towards studying organisational functions of volunteering systematically, future research should also further investigate the complex organisational frameworks in which such projects take place e.g., [3].

When investigating motivations to participate in a citizen science project about rainwater in Mexico, the three top reasons to participate were project-specific (e.g., positive attitude towards project purpose). Furthermore, the study found considerable differences in motivations for participants with different personal characteristics (e.g., education, community membership) [50]. Consequently, specific characteristics of citizen science projects and CSs should always be considered. Regarding the personalities of CSs, it would be interesting to further investigate to what degree *nature conservation* is part of their identity. Given the established community of *birders* with their distinct identity, which differs from that of environmentalists [51], we might also find an identity of *insecters*? Udall et al. published in 2020 a theoretical framework of how to test identities in relation to other psychological variables relevant for pro-environmental behaviour [52]. In addition, distinguishing between volunteers who prefer to work and learn autonomously and volunteers who prefer to follow instructions could be interesting for future research.

Last, but not least, we derived recommendations to optimise the volunteer management in the project *Insects of Saxony*. As we saw in the second group of our Study 4, where volunteers explore the effects of insect-friendly habitats on diversity and which is coordinated by the same project manager, the ratio of men (42%) and women (46%) is much more balanced. From this we conclude that there was no (unintentional) exclusion of women and that they obviously choose the project that they like better. However, the CSs’ personal contribution to the research should be emphasised and the collaboration with the institutional scientists improved, to prevent drop out due to dissatisfaction [42]. To tap into the full potential of citizen science, the project could find ways that attract also non-academics and youngsters.

## Figures and Tables

**Figure 1 insects-12-00262-f001:**
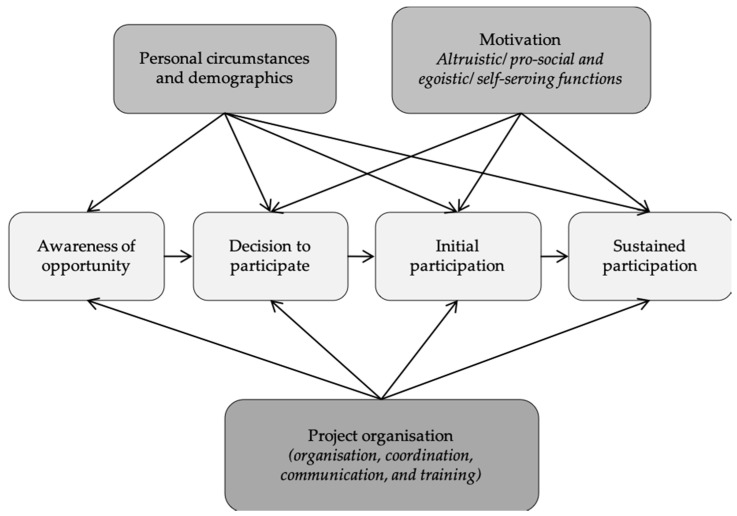
Model of Influence for participation in citizen science projects by Penner (2002) [17], adapted by and cited from West et al. (2016) in Geoghegan et al. (2016) [11]. *Additions* by Moczek, 2019 [18].

**Figure 2 insects-12-00262-f002:**
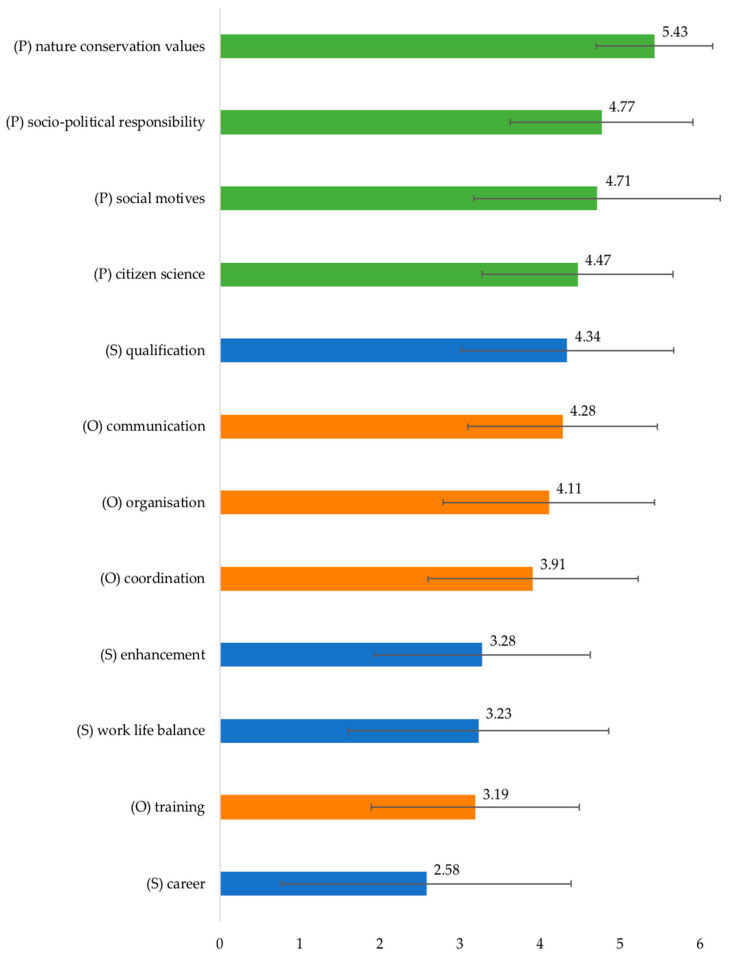
Functions for participating in the project Insects of Saxony, *N* = 116. Six-point scale: 1 = does not apply at all 6 = fully applies; mean values (coloured bars) and standard deviations (error bars). In descending order: (P) (green bars) = pro-social functions, (S) (blue bars) = self-serving functions, (O) (orange bars) = organisational functions.

**Figure 3 insects-12-00262-f003:**
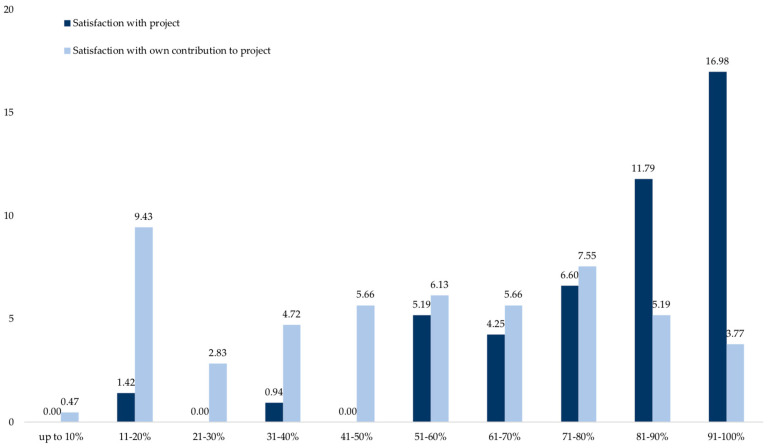
Satisfaction with the overall project and personal contribution to project in percent (*N* = 115). Eleven-point scale from 0 = 0%, 1 = up to 10%, to 11 = 91–100%; mean values.

**Table 1 insects-12-00262-t001:** Scientific activities of the citizen scientists ^1^.

Scientific Activities	Never	Occasionally	Often	Mean	SD
Observing (e.g., phenomena, species identification, photography, audio)	5.4	49.1	45.5	2.4	0.59
Reporting observations (on the online database)	6.0	65.5	28.4	2.22	0.55
Implementation (care for meadows or conservation areas)	49.5	38.7	11.7	1.62	0.69
Communicating (e.g., with authorities, politics, public, press)	50.9	43.6	5.5	1.55	0.6
Measuring (e.g., counting, collecting data)	56.0	29.4	14.7	1.59	0.74
Formulating new research questions or topics/draw attention to a phenomenon	59.6	38.5	1.8	1.42	0.53
Analysis (e.g., data)	71.3	21.3	7.4	1.36	0.62
Quality checking of records	75.0	15.7	9.3	1.34	0.64

^1^ Note: The eight items were answered by *N* = 108–111 participants. Data for never, occasionally and often in percent. Data sorted in ascending order of answer never.

**Table 2 insects-12-00262-t002:** Items and item analysis of Motivational and ORganisational Functions of voluntary ENgagement in Citizen Science (MORFEN-CS) ^1^.

Item		*M*	*SD*	P_i_	r_it_	Alpha *Alpha of Subfactor If Item Is Deleted*
*No.*	*I am volunteering in this project, because (I)…*					
	**Motivational Functions (pro-social *)**	**4.71**	**1.54**			**0.654**
	**Nature Conservation Values**	**5.43**	**0.73**			**0.814**
1	…can do something for a cause that is personally important to me.	5.54	0.93	90.80	0.62	*0.774*
2	…my personal values match the project goals.	5.19	1.16	83.89	0.66	*0.764*
3	…can actively contribute to nature conservation in the project.	5.29	1.06	85.84	0.62	*0.774*
4	…like to support the preservation of wildlife.	5.45	0.90	89.03	0.54	*0.797*
5	… want to do something to help stop the loss of habitats.	5.63	0.69	92.63	0.52	*0.778*
	**Social Motives**	**4.71**	**1.54**			**0.903**
6	…am part of a community supporting the same cause.	4.13	1.66	62.65	0.82	*0.843*
7	…can get involved together with others.	3.90	1.61	58.07	0.85	*0.816*
8	…meet people with similar interests.	3.80	1.68	55.96	0.72	*0.923*
	**Socio-Political Responsibility**	**4.77**	**1.14**			**0.676**
9	…like to rectify deficits in nature conservation.	4.96	1.27	79.12	0.40	*0.679*
10	…like to perform a socially meaningful task.	4.45	1.59	68.95	0.41	*0.668*
11	…like to initiate political changes concerning nature conservation.	4.81	1.49	76.17	0.62	*0.343*
	**Citizen Science**	**4.47**	**1.19**			**0.824**
12	…want to support a scientific research project.	4.55	1.61	70.97	0.72	*0.756*
13	…am interested in a professional exchange with scientists in the project.	3.88	1.71	57.54	0.70	*0.752*
14	…can learn to understand scientific processes better.	3.64	1.78	52.74	0.63	*0.785*
15	…want to contribute to species identification and environmental monitoring.	5.61	0.79	92.17	0.37	*0.842*
16	…can engage in knowledge exchange among citizens and scientists.	4.54	1.57	70.78	0.57	*0.784*
	**Motivational functions (self-serving *)**	**3.36**	**1.89**			**0.600**
	**Qualification**	**4.34**	**1.33**			**0.786**
17	…can gain new perspectives on nature.	4.09	1.50	61.90	0.34	*0.870*
18	…can learn something new and apply it through my concrete actions.	4.66	1.41	73.22	0.36	*0.317*
19	…can learn and apply theoretical knowledge and methods.	4.25	1.56	65.09	0.26	*0.838*
	**Enhancement**	**3.28**	**1.35**			**0.755**
20	…get the impression of being needed.	3.59	1.59	51.83	0.66	*0.592*
21	…receive recognition for my contribution.	2.62	1.54	32.35	0.82	*0.652*
22	…can self-realise myself.	3.63	1.79	52.70	0.87	*0.772*
	**Work Life Balance**	**3.23**	**1.63**			**0.800**
23	…can do everything I want in my volunteering-unlike in professional life.	3.31	1.88	46.26	0.46	*0.857*
24	…find a meaningful balance to my professional job.	2.74	1.88	34.87	0.93	*0.613*
25	…can recover from job requirements by being in nature.	3.56	1.97	51.25	0.94	*0.685*
	**Career**	**2.58**	**1.81**			**0.922**
26	…like to gain experience that I can also use in my job.	2.82	1.97	36.46	0.81	*0.917*
27	…volunteering might positively affect my professional skills.	2.57	1.95	31.33	0.89	*0.856*
28	…can establish and cultivate contacts that can be beneficial for my career.	2.36	1.89	27.26	0.85	*0.889*
	**Organisational functions**					
	**Training**	**3.19**	**1.30**			**0.817**
29	…know which tasks I can perform in the project.	3.48	1.62	49.55	0.61	*0.815*
30	…am getting an introduction into scientific methods.	2.99	1.50	39.82	0.70	*0.722*
31	…can work with scientific methods.	3.09	1.44	41.80	0.72	*0.710*
	**Coordination**	**3.91**	**1.31**			**0.810**
32	…determine time and duration of my engagement myself.	4.93	1.35	78.56	0.40	*0.843*
33	…there is regular contact with the project staff.	3.05	1.75	40.91	0.61	*0.754*
34	…can choose between different tasks and actions in the project.	3.66	1.80	53.27	0.77	*0.694*
35	…am experiencing good support overall.	3.89	1.55	57.84	0.67	*0.727*
	**Communication and Feedback**	**4.28**	**1.18**			**0.752**
36	…promptly get feedback on the results of my work.	3.94	1.59	58.77	0.47	*0.727*
37	…am given information on successes in the overall project.	4.07	1.47	61.42	0.71	*0.516*
38	…get the impression that my personal engagement is helpful for the entire project.	4.81	1.23	76.11	0.53	*0.736*
	**Organisation**	**4.11**	**1.32**			**0.842**
39	…the project is very well organised overall.	4.62	1.41	72.39	0.69	*0.783*
40	…work materials are provided.	3.43	1.77	48.65	0.67	*0.813*
41	…the overall project goal is clear to me.	4.77	1.48	75.44	0.70	*0.773*
42	…the project is carried by a society/organisation.	3.54	1.70	50.81	0.60	*0.832*

^1^ Notes: *N* = 112–116. Six-point scale: 1 = does not apply at all 6 = fully applies; intermediate steps were not verbally anchored and the headings were not presented. Pi: Item difficulty. Value range between 0 and 100. The higher the value, the easier it is on average for the respondents to give an affirmative answer to the item. Values between 20 and 80 are preferred. R_it_: Item-total correlation or Item selectivity. Correlative relationship between an individual item and the overall test. Value range between 0 and 1. The selectivity is intended to enable an assessment of how well an item distinguishes between people with low and high characteristics. Values between 0.4 and 0.7 are preferred. A high level of item variance favours a high degree of selectivity. Alpha: Tau-equivalent reliability also known as Cronbach’s alpha or coefficient alpha. Value range between 0 and 1. The higher the value the better the internal consistency. When used for collective diagnostics, as in the present case, values of around 0.7 are acceptable. Alpha of subfactor if item is deleted (in italics). * The confirmatory factor analyses calculated with this data did not confirm the two superior factors pro-social/altruistic and self-serving/egoistic.

**Table 3 insects-12-00262-t003:** Self-assessment of the level of expertise prior to the commitment and at the time of the survey ^1^.

Level of Expertise	*M*	*SD*	Laity	Low Expertise	Medium Expertise	Extensive Expertise	Expert
Prior to commitment	2.27	1.15	36.92	33.04	25.22	27.83	9.57
at the time of the survey	2.83	1.06	9.81	9.57	28.70	38.26	15.65
difference	0.56	0.69	−27.10	−23.48	3.48	10.43	6.09

^1^ Notes: *N* = 114. Numbers in percent.

**Table 4 insects-12-00262-t004:** Intended participation in 2019 ^1^.

Activity	Much Less			Much More			No Change to 2018
	1	2	3	4	5	6	−1
invest time in the project	0.0	0.0	3.5	32.7	23.9	11.5	28.3
learn about specific insects	0.0	0.0	4.4	18.6	33.6	27.4	15.9
learn about different insect species	0.0	0.9	3.5	16.8	36.3	27.4	15.0
learn about habitats	0.0	0.0	1.8	16.8	37.2	28.3	15.9
contribute to the protection of insects	0.0	0.9	0.9	23.9	25.7	29.2	19.5

^1^ Notes: *N* = 113. Numbers in percent. Six-point scale: 1–3 = *much less*, 4–6 *much more*; −1 *no change to 2018*.

## Data Availability

The data presented in this study are available on request from the corresponding author. The data are not publicly available due to only an excerpt of the data being presented in this article and further publications planned.

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
