# Peer review of "Volunteering in the Citizen Science Project “Insects of Saxony”—The Larger the Island of Knowledge, the Longer the Bank of Questions"

_insects, 2021, doi:10.3390/insects12030262_

Round 1

Reviewer 1 Report

This paper presents data that describe the motivations of citizen scientists who contribute data to a research project in Germany, using a newly developed measurement tool (MORFEN-CS) that is specific to volunteers in ecological citizen science research.  The study is well done, and the paper well written.  My recommendations for revision focus on two issues: more thoroughly describing statistical methods that may not be familiar to scientists outside of this field of research, and elaboration on the potential for the self-selection of study participants to influence conclusions about volunteer motivation. 

Substantive questions:

  1. I study insect ecology and behavior, so the analyses used in this manuscript are not ones that I’m familiar with. Since this manuscript has been submitted to the journal “Insects” rather than to a journal more specifically focused on measuring human motivation, this is likely to describe much of the target audience.  I found this study interesting, and I think many readers of this journal will also be interested – citizen science research has become a valuable way to efficiently collect data.  However, because the target audience may be unfamiliar with these statistics, you might consider explaining them in a bit more detail.  In Lines 292-293, you explained Pi very clearly – I don’t need to know how it was calculated, but this sentence was enough for me to understand that high numbers indicate how easy the question was to answer – that helped me follow that part of the table.  I was still wondering about the range of Pi values – do possible values range from 0-100?  Is this calculated based on how many of the respondents answered this question?  The explanation of rit was less clear – I understand that you wanted the values to fall between 0.4-0.7, and that they all did, and that this has something to do with being able to differentiate between subjects.  So, I’m guessing that this means that there’s enough variation to distinguish between subjects, and perhaps has something to do with internal consistency within/between respondents, but beyond that I didn’t see anything else in the methods or elsewhere in the paper that provided further clarification.  Are both high and low values of rit problematic?  Do the values range from 0-1?  I was also unclear about the implication of Alpha.  I’m used to seeing alpha with reference to p-values, but that doesn’t fit in this content, and you’ve defined alpha in the notes that follow Table 2, as the tau equivalent reliability, if item is deleted.  But without further explanation in the methods, I’m not sure what that means.  I quick search for “tau equivalent reliability” tells me that this relates to test score reliability, which makes me think that this may be a statistic that would be familiar to researchers who study assessments of human behavior, but again – I don’t think that describes the majority of the audience of the journal “Insects” so you might consider adding a bit more explanation to help out your readers who, like me, are interested in your research, but not familiar with the methods commonly used to assess volunteer motivations.  I have been involved with multiple citizen science projects over the years, so I find this study very interesting, but a bit more explanation of the statistics used would help readers like me.
  2. Figure 2: It’s not clear to me why you have plotted standard deviations as separate bars next to the bars that show the mean values. Since standard deviation is a measure of variability around (+/-) the mean, I have always seen that plotted as error bars that show variation above and below the mean.  I’m trying to think of a reason why it would make sense to plot this measure of variability as a separate bar, and I can’t think of one.  So, it seems to me that standard deviation should be plotted as error bars rather than as separate bars.
  3. Figures 2 and 3 – do the numbers associated with each bar (e.g. 0,73) just indicate the value that is plotted on the bar? I was confused by this at first, because I’m from the U.S., where commas are used to separate two different numbers, whereas decimal places are separated from the integer by a period/dot.  So, when I read this graph, I thought that the numbers next to each bar indicated 2 separate values, but I couldn’t figure out which values (I thought one might have been the sample size, but that didn’t make sense…).  Then I remembered that in other countries, decimal places are separated by commas, and that fits in this context.  But, since there is the potential for there to be this confusion depending on the nationality of the reader, you might specify in the figure titles what those numbers show (although if they do just show the value indicated by the bar, I think these numbers are unnecessary and distracting – just let the bars do their job, but I recognize that this is a personal preference so you’re welcome to disagree with me and leave them on).
  4. Lines 440-448: I appreciate that you have acknowledged some limitations of this study, but I think you should develop this section a bit further, since this may affect your conclusions. Lines 23-24 (abstract) list as one of your key conclusions that pro-social functions were more highly rated by the volunteers as important for their engagement.  However, the act of responding to this survey was itself pro-social, which means that of the 202 volunteers who were sent the survey, the ~50% who responded were probably more likely to be driven by pro-social motivations.  Any CSs who are more driven by self-serving motivations are arguably less likely to respond to this kind of a survey, which creates a bias in your respondent pool.  I don’t think this negates the value of this study or this manuscript, but I think this should be acknowledged in this section about limitations more explicitly than just the reference in Line 442 to a risk of self-selection bias. 
  5. Supporting Figure 1 is really great – I would love to see it include a reference to the categories of motivations (e.g. O/P/S), either with letters included after the question wording (e.g. “…want to do something to help stop the loss of habitats. (P)”) or by color coding the bars as you did in Figure 2.

Editorial comments:

  1. Line 14: “applied” instead of “applicated”
  2. Lines 37-40: might be good to clarify the geographic scope of these statistics – I’m guessing this refers to within Germany (the numbers seem to small to be global), but since this journal is not specific to Germany, it would be good to clarify this.
  3. Line 46: you might specify the acronym here (CBD) since you use it again 2 lines later
  4. Line 64: extant seems an odd word choice here, since it refers to being still in existence as opposed to others that have disappeared, but this sentence refers to data that are just beginning to exist, so… “available” might be a better word choice
  5. Line 89: “affect” should be “effect”
  6. Line 110: since data is the plural of datum, “is” should be “are” (or if you don’t like the sound of that, you could rearrange as “Where data exist…”)
  7. Line 137: is this acronym wrong – seems like it should be CTT, unless this is an issue that results from translation of “classical test theory” from another language?
  8. Line 437: “were” should be “where”
  9. Line 464: “specially” should be “especially”

Author Response

We have summarised the detailed comments of the reviewers in one document. 

Reviewer 2 Report

Authors,

Thank you for your interesting submission entitled “Volunteering in the Citizen Science Project “Insects of Saxony”. The larger the island of knowledge, the longer the bank of questions.” This research seeks to characterize the aspects of project design and internal/external motivators for individuals to participate in an insect-specific citizen science project. This research has findings similar to other research in terms of citizen science demographics and other identity metrics. It would be more interesting to see the authors parse out the data from the minority demographic participants (i.e., women, lower education brackets) and compare your measures of project motivation/enjoyment/knowledge from these groups to the majority demographic responses. For example, do women, who are a minority group in this citizen science project, align with men in terms of their motivations to contribute, or do you see divergence? I am concerned because you present this insect citizen scientist identity as a monolith where I’m not convinced by the data and descriptive statistics you present here that this is true.

For your methods, can you clarify what you mean by contentment? I see on line 237-238 you have it on an 11 point scale, but some clarity around what this means would be helpful. Specifically, what exactly is this a measure of? What does this measure tell us about the participants? I suspect there might not be ubiquitous understanding of what a 9 out of 11 means in terms of contentment and further what contentment is in terms of a citizen scientists psycho-social-emotional capacity around a project.

Looking at your results, section 3.3 for organizational structures, it is interesting that item 15 is such an incredibly strong driver above and beyond other items even those arguably in terms of meaningful impact on scientific process and real-world outcomes many of the other items are more important. This results really speaks to having practitioners think about designing projects that motivate individuals, in this case for insect citizen science to engage around identifying species and developing skills in this area versus other more altruistic drivers. I also found it interesting that items 30 and 31 suggest that, for at least your participants, they are not interested in engaging in the genuine process of science. These two findings combined seem to lend support that as citizen science researchers, we need to be careful in conflating data contribution with genuine engagement in learning about the enterprise of science.

All in all, I think there are a number of potential interesting take-aways here for citizen science practitioners and researchers, however I think this paper could also be more impactful with another pass at evaluating the data you have.

Author Response

(The authors gave the same response as above.)

Author Response

(The authors gave the same response as above.)
